# Efficacy and safety of bempedoic acid for the treatment of hypercholesterolemia: A systematic review and meta-analysis

Arrigo F. G. Cicero[1]*, Federica Fogacci[1], Adrian V. Hernandez[2,3], Maciej Banach[4,5,6]*, on behalf of the Lipid and Blood Pressure Meta-Analysis Collaboration (LBPMC) Group and the International Lipid Expert Panel (ILEP)[¶]

1 Hypertension and Cardiovascular Risk Factors Research Group, Department of Medicine and Surgery Sciences, University of Bologna, Bologna, Italy, 2 Health Outcomes, Policy, and Evidence Synthesis (HOPES) Group, University of Connecticut/Hartford Hospital Evidence-based Practice Center, Hartford, Connecticut, United States of America, 3 Vicerrectorado de Investigacion, Universidad San Ignacio de Loyola, Lima, Peru, 4 Chair of Nephrology and Hypertension, Department of Hypertension, Medical University of Lodz, Lodz, Poland, 5 Polish Mother's Memorial Hospital Research Institute, Lodz, Poland, 6 Cardiovascular Research Centre, University of Zielona Gora, Zielona Gora, Poland

¶ Membership of the International Lipid Expert Panel (ILEP) is provided in the Acknowledgments.
* arrigo.cicero@unibo.it (AFGC); maciejbanach77@gmail.com (MB)

## Abstract

### Background

Bempedoic acid is a first-in-class lipid-lowering drug recommended by guidelines for the treatment of hypercholesterolemia. Our objective was to estimate its average effect on plasma lipids in humans and its safety profile.

### Methods and findings

We carried out a systematic review and meta-analysis of phase II and III randomized controlled trials on bempedoic acid (PROSPERO: CRD42019129687). PubMed (Medline), Scopus, Google Scholar, and Web of Science databases were searched, with no language restriction, from inception to 5 August 2019. We included 10 RCTs ($n = 3,788$) comprising 26 arms (active arm [$n = 2,460$]; control arm [$n = 1,328$]). Effect sizes for changes in lipids and high-sensitivity C-reactive protein (hsCRP) serum concentration were expressed as mean differences (MDs) and 95% confidence intervals (CIs). For safety analyses, odds ratios (ORs) and 95% CIs were calculated using the Mantel–Haenszel method. Bempedoic acid significantly reduced total cholesterol (MD −14.94%; 95% CI −17.31%, −12.57%; $p <$ 0.001), non-high-density lipoprotein cholesterol (MD −18.17%; 95% CI −21.14%, −15.19%; $p <$ 0.001), low-density lipoprotein cholesterol (MD −22.94%; 95% CI −26.63%, −19.25%; $p <$ 0.001), low-density lipoprotein particle number (MD −20.67%; 95% CI −23.84%, −17.48%; $p <$ 0.001), apolipoprotein B (MD −15.18%; 95% CI −17.41%, −12.95%; $p <$ 0.001), high-density lipoprotein cholesterol (MD −5.83%; 95% CI −6.14%, −5.52%; $p <$ 0.001), high-density lipoprotein particle number (MD −3.21%; 95% CI −6.40%, −0.02%; $p =$ 0.049), and hsCRP (MD −27.03%; 95% CI −31.42%, −22.64%; $p <$ 0.001). Bempedoic acid did not

**Data Availability Statement:** All relevant data are within the manuscript and its Supporting Information files.

**Funding:** The authors received no specific funding for this work.

**Competing interests:** I have read the journal's policy and the authors of this manuscript have the following competing interests: MB has received research grant(s)/support from Sanofi and Valeant, and has served as a consultant for Abbott/Mylan, Akcea, Amgen, KRKA, MSD, Polfarmex, Polpharma, Sanofi-Aventis/Regeneron, Servier, Esperion, and Resverlogix. AFGC was scientific consultant for Mylan and Menarini International. FF was scientific consultant for Mylan. AH has no conflict of interest to declare.

**Abbreviations:** ACLY, ATP citrate lyase; AE, adverse event; Apo, apolipoprotein; CI, confidence interval; CV, cardiovascular; CVD, cardiovascular disease; HDL-C, high-density lipoprotein cholesterol; hsCRP, high-sensitivity C-reactive protein; LDL-C, low-density lipoprotein cholesterol; MD, mean difference; OR, odds ratio; RCT, randomized controlled trial; SD, standard deviation; SUA, serum uric acid; TC, total cholesterol; TG, triglyceride; VLDL, very-low-density lipoprotein.

significantly modify triglyceride level (MD −1.51%; 95% CI −3.75%, 0.74%; $p = 0.189$), very-low-density lipoprotein particle number (MD 3.79%; 95% CI −9.81%, 17.39%; $p = 0.585$), and apolipoprotein A-1 (MD −1.83%; 95% CI −5.23%, 1.56%; $p = 0.290$). Treatment with bempedoic acid was positively associated with an increased risk of discontinuation of treatment (OR 1.37; 95% CI 1.06, 1.76; $p = 0.015$), elevated serum uric acid (OR 3.55; 95% CI 1.03, 12.27; $p = 0.045$), elevated liver enzymes (OR 4.28; 95% CI 1.34, 13.71; $p = 0.014$), and elevated creatine kinase (OR 3.79; 95% CI 1.06, 13.51; $p = 0.04$), though it was strongly associated with a decreased risk of new onset or worsening diabetes (OR 0.59; 95% CI 0.39, 0.90; $p = 0.01$). The main limitation of this meta-analysis is related to the relatively small number of individuals involved in the studies, which were often short or middle term in length.

## Conclusions

Our results show that bempedoic acid has favorable effects on lipid profile and hsCRP levels and an acceptable safety profile. Further well-designed studies are needed to explore its longer-term safety.

## Author summary

### Why was this study done?

- Lowering low-density lipoprotein cholesterol (LDL-C) is effective for reducing cardiovascular events over time.

- A number of phase II and phase III randomized controlled trials (RCTs) are already available showing encouraging results of bempedoic acid treatment on LDL-C.

- We aimed to perform a systematic review and meta-analysis on the clinical evidence available to date to better define the efficacy and tolerability profile of treatment with bempedoic acid.

### What did the researchers do and find?

- In this analysis of bempedoic acid that included 10 randomized clinical trials ($n = 3,788$ patients) comprising 26 arms (active arm [$n = 2,460$]; control arm [$n = 1,328$]), we confirmed that bempedoic acid significantly reduced total cholesterol (by 15%), non-high-density lipoprotein cholesterol (by 18.2%), LDL-C (by 22.9%), low-density lipoprotein particle number (by 20.7%), apolipoprotein B (by 15.2%), and high-sensitivity C-reactive protein (hsCRP) (by 27%), while negatively affecting serum levels of high-density lipoprotein cholesterol (−5.8%) and high-density lipoprotein particle number (−3.2%).

- Our results also confirmed that the therapy is overall safe and well tolerated, with no significant increase of serious adverse effects.

**What do these findings mean?**

- The current meta-analysis demonstrates the multiple positive effects of bempedoic acid on lipid profile and hsCRP serum levels, as well as acceptable safety profile.

- This could be relevant in a setting where statin intolerance is very frequent and the LDL-C target suggested by international guidelines for dyslipidemia management is hard to achieve with standard therapies.

- An ongoing long-term cardiovascular outcomes trial will answer questions on the effect of bempedoic acid on cardiovascular events and mortality as well as on the drug's safety issues.

## Introduction

Cardiovascular diseases (CVDs) are still the leading cause of disability and death in developed countries [1]. As reported by Mendelian randomization studies, a lifetime reduction of low-density lipoprotein cholesterol (LDL-C) of 1 mmol/l might reduce the potential risk of athero-sclerotic CVDs by over 50% [2]. Controlled clinical studies successfully showed a consistent relationship between the reduction of LDL-C and cardiovascular (CV) risk decrease [3], such that lipid-lowering therapy became a cornerstone in CV risk reduction.

Bempedoic acid (8-hydroxy-2,2,14,14-tetramethylpentadecanedioic acid; ETC-1002; Esperion Therapeutics, Ann Arbor, MI) is a first-in-class small-molecule inhibitor of ATP citrate lyase (ACLY), a key enzyme that supplies substrate for cholesterol and fatty acid synthesis [4]. ACLY is essential for growth and development, such that homozygous knockout (Acly$^-$) in mice is embryonic lethal, indicating non-redundancy during development [5]. By inhibiting ACLY, bempedoic acid induces LDL receptor upregulation and stimulates the uptake of LDL particles by the liver, which contributes to reduction of LDL-C concentration in the blood [6]. Bempedoic acid is administered orally once a day, is quickly absorbed in the small intestine, and has a half-life ranging from 15 to 24 hours [7]. It is a prodrug that is activated by very-long-chain acyl-CoA synthetase 1, an enzyme that is synthesized only in the liver [8]. Even though bempedoic acid acts on the same pathway as statins (3-hydroxy-3-methylglutaryl coen-zyme A reductase inhibitors), the lack of the activating enzyme in skeletal muscle may prevent the muscular adverse effects associated with statins [8]. For this reason, bempedoic acid may represent a novel treatment to reach LDL-C goals for statin-intolerant patients [9].

A number of phase II and phase III randomized controlled trials (RCTs) are already available, showing encouraging effects of bempedoic acid treatment on LDL-C. Consequently, we aimed to perform a systematic review and meta-analysis of the clinical evidence available to date to better define its efficacy and tolerability profile.

## Methods

The study is reported in accordance with the 2009 guidelines of the Preferred Reporting Items for Systematic Reviews and Meta-analyses (PRISMA) statement (S1 PRISMA Checklist) [10], and was registered in the PROSPERO database (registration code: CRD42019129687). Due to the study design (meta-analysis), neither institutional review board approval nor patient informed consent was required.

## Search strategy

PubMed (Medline), Web of Science, Google Scholar, and Scopus databases were searched, with no language restriction, using the following search terms: ("Bempedoic acid" OR "ETC-1002") AND ("Trial" OR "Study") [Search terms: (("Bempedoic acid") AND Study) OR ((Bempedoic acid) AND Trial) OR (ETC-1002 AND Study) OR (ETC-1002 AND Trial))]. The wild-card term "*" was used to increase the sensitivity of the search strategy, which was limited to studies in humans. The reference lists of identified papers were manually checked for additional relevant articles. Additional searches for potential trials included the references of review articles on bempedoic acid, and the abstracts from selected scientific conferences on the subject of the meta-analysis. Literature was searched from inception to 5 August 2019.

All abstracts were screened by 2 reviewers (FF and AFGC) in an initial process to remove ineligible articles. The remaining articles were obtained in full-text and assessed again by the same 2 researchers, who evaluated each article independently and carried out data extraction and quality assessment. Disagreements were resolved by discussion with a third party (MB).

## Study selection criteria

Original studies were included if they met the following criteria: (i) were a phase II or III RCT with either multicenter or single-center design, (ii) investigated the effect of bempedoic acid on plasma lipids or high-sensitivity C-reactive protein (hsCRP), (iii) tested the safety of bempedoic acid in short- and middle-term administration, and (iv) reported all the adverse events (AEs) that occurred during the treatment.

Studies that lacked a control-treated group for comparison with bempedoic acid were excluded.

## Data extraction

Data abstracted from the eligible studies were the following: (i) study registration code; (ii) first author's name; (iii) publication year; (iv) study phase; (v) main inclusion criteria and underlying disease; (vi) treatment duration; (vii) study arms; (viii) number of participants in the active and control group; (ix) age and sex of study participants, (x) baseline and outcome data of total cholesterol (TC), LDL-C, high-density lipoprotein cholesterol (HDL-C), very-low-density lipoprotein (VLDL), non-HDL-C, triglycerides (TGs), apolipoprotein (Apo) B, Apo A-1, and hsCRP; and (xi) discontinuation of treatment and AEs that occurred during the trials. Safety outcomes included: AEs, serious AEs, study-drug-related AEs, AEs leading to discontinuation of treatment, death, major adverse cardiac events, muscle-related AEs, arthralgia, gout, back pain, pain in extremity, pruritus, rash, new onset hypertension, headache, fatigue, dizziness, dyspepsia, abdominal pain, nausea, constipation, diarrhea, nasopharyngitis, sinusitis, cough, dyspnea, upper respiratory tract infection, bronchitis, urinary tract infection, vulvovaginal mycotic infection, new onset or worsening diabetes, neurocognitive disorders, vertigo, increase in blood creatinine level, decrease in glomerular filtration rate, creatine kinase (CK) elevation serum uric acid (SUA) elevation, and liver enzyme (transaminase and gamma-glutamyl transferase) elevation. All the verbatim terms for the AEs were coded to preferred term and System Organ Class with the use of the Medical Dictionary for Regulatory Activities (MedDRA).

Missing or unpublished data were sought by trying to contact authors or sponsors via e-mail, and, in cases of no response, repeated messages were sent. Data extraction and database typing were performed by 2 authors (AFGC and FF) and reviewed by a third author (MB) before the final analysis. Doubts were resolved by mutual agreement among the authors.

### Risk of bias evaluation

A systematic evaluation of risk of bias in the included studies was performed using the Cochrane tool [11]. The items used were the following: adequacy of sequence generation, blinding, addressing of dropouts (incomplete outcome data), allocation concealment, selective outcome reporting, and other probable sources of bias [12]. Risk of bias assessment was performed by 2 reviewers (FF and AFGC) independently; disagreements were resolved by a consensus-based discussion. Each item was judged as high, low, or unclear risk of bias. A trial with high risk of bias in the randomization or blinding items was judged as having high risk of bias overall.

### Data synthesis

All analyses were performed with Comprehensive Meta-Analysis (CMA) version 3 software (Biostat, Englewood, NJ) [13]. Changes in continuous outcomes were calculated for each study arm by subtracting the value at baseline from the one after intervention. All values were expressed as percent change from baseline. Standard deviations (SDs) of the mean differences (MDs) were obtained as follows, per Follmann and colleagues [14]: $SD = \sqrt{[SD_{pre}^2 + SD_{post}^2 - (2R \times SD_{pre} \times SD_{post})]}$, assuming a correlation coefficient ($R$) of 0.5. If the outcome measures were reported in the original articles as median and interquartile range (or 95% confidence interval [CI]), mean and SD values were obtained as described by Wan et al. [15]. In case standard error of the mean (SEM) was only reported as a dispersion measure, SD was estimated using the following formula: $SD = SEM \times \sqrt{n}$, with $n$ being the number of individuals. To handle the double-counting problem in trials comparing different treatments against a single control group, individuals within the control group were divided by the required comparisons.

Meta-analyses were conducted using a fixed-effect model or a random-effect model (using the DerSimonian–Laird method) and the generic inverse variance method based on the moderately low (<50%) or high (≥50%) inter-study heterogeneity, which was quantitatively assessed using the Higgins index ($I^2$) [16]. Effect sizes for lipid and hsCRP changes were expressed as MDs and 95% CIs. For safety analyses, odds ratios (ORs) and 95% CIs were calculated using the Mantel–Haenszel method [17]. If 1 or more outcomes could not be extracted from a study, the study was removed from the analysis involving those outcomes. AEs were included in the analysis only if occurring in at least 2 of the selected clinical trials. The efficacy analysis was performed on the safety population; the analysis of safety data was based on the intention-to-treat population.

For the purpose of evaluating the influence of each study on the overall effect size, sensitivity analysis was conducted using the leave-one-out method (i.e., repeating the analysis after omitting 1 study at a time) [18]. Two-sided $p$-values ≤ 0.05 were considered statistically significant for all tests.

If statistical heterogeneity was detected, attempts to identify the sources of heterogeneity and potential publication biases were made through the visual inspection of Begg's funnel plot asymmetry, and carrying out the Begg's rank correlation test and Egger's linear regression test [19]. The Duval and Tweedie "trim and fill" method was used to adjust the analysis for the effects of publication bias [20]. In case of a significant result, the number of potentially missing studies required to make the $p$-value non-significant was estimated by using the classical fail-safe $N$ method as another marker of publication bias. Two-sided $p$-values ≤ 0.05 were considered statistically significant.

## Results

### Flow and characteristics of the included studies

We identified 248 published abstracts. Of these, 238 were excluded because they were not original articles. All the other 10 studies met the inclusion criteria and were carefully assessed and

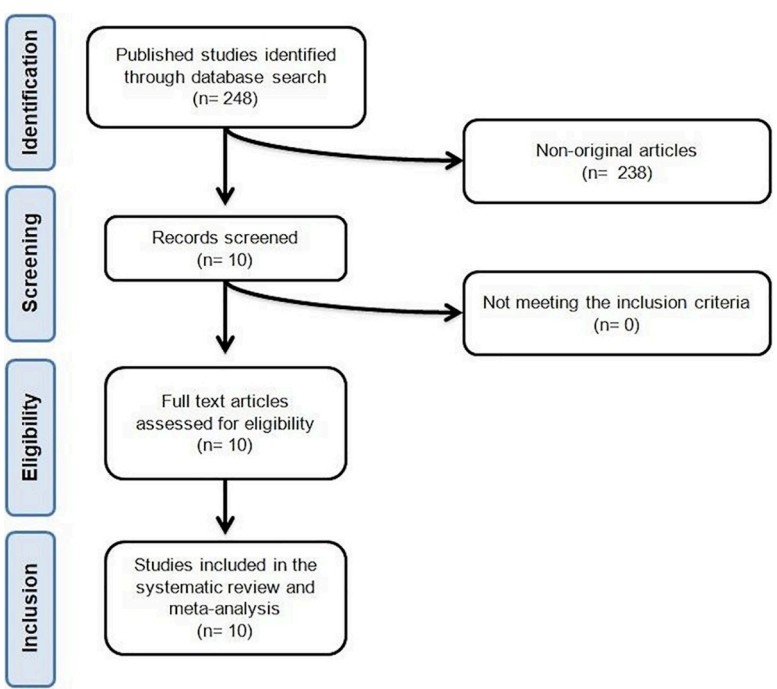

**Fig 1. Flow chart of the number of studies identified and included in the meta-analysis.** Data were pooled from 10 trials comprising 26 treatment arms, which included overall 3,788 individuals, with 2,460 in the active arm and 1,328 in the control arm.

reviewed. On the basis of the established eligibility criteria, all 10 RCTs were included in the meta-analysis [9,21–29]. The study selection process is shown in Fig 1.

Eligible studies were published between 2013 and 2019. Follow-up periods ranged between 4 and 52 weeks, and several treatment schedules were tested. All trials were parallel and multi-center [9,21–26,28,29] or single-center [27]. Enrolled individuals were statin-intolerant individuals [9,21,24,28], patients with type 2 diabetes [21,27], or patients affected by hypercholesterolemia despite statin treatment [21–23,25,26,29]. The main characteristics of the selected studies are summarized in Table 1.

## Risk of bias evaluation

The studies reported sufficient information regarding sequence generation, allocation concealment, blinding of participants, personnel, and outcome assessment. Details of the risk of bias evaluation are reported in Table 2.

## Effect of bempedoic acid on selected laboratory parameters

Meta-analysis of available data showed that bempedoic acid significantly reduced TC ($n$ = 3,485; MD −14.94%; 95% CI −17.31%, −12.57%; $p < 0.001$; $I^2$ = 76.1%) (Fig 2), non-HDL-C ($n$ = 3,485; MD −18.17%; 95% CI −21.14%, −15.19%; $p < 0.001$; $I^2$ = 87.2%) (Fig 3), LDL-C ($n$ = 3,483; MD −22.94%; 95% CI −26.63%, −19.25%; $p < 0.001$; $I^2$ = 77.3%) (Fig 4), LDL particle number ($n$ = 441; MD −20.67%; 95% CI −23.84%, −17.48%; $p < 0.001$; $I^2$ = 0%) (Fig 5), Apo B ($n$ = 3,402; MD −15.18%; 95% CI −17.41%, −12.95%; $p < 0.001$; $I^2$ = 81.4%) (Fig 6), HDL-C ($n$ = 3,453; MD −5.83%; 95% CI −6.14%, −5.52%; $p < 0.001$; $I^2$ = 33.4%) (Fig 7), and hsCRP ($n$ = 3,179; MD −27.03%; 95% CI −31.42%, −22.64%; $p < 0.001$; $I^2$ = 0%) (Fig 8). Furthermore,

**Table 1. Main characteristics of the selected studies.**

| Study | First author, year [reference] | Study design | Main inclusion criteria | Primary outcomes | Treatment duration | Study groups | Patients, *n* | Age (years), mean ± SD | Female, *n* (%) | Average change in LDL-C from baseline |
|---|---|---|---|---|---|---|---|---|---|---|
| NCT03337308 | Ballantyne, 2019 [21] | Multicenter, randomized, double-blind, placebo-controlled, parallel-group, phase III clinical study | ≥18 years of age; high risk for CVD; LDL-C ≥ 2.4 mmol/l for ASCVD or HeFH patients and LDL-C ≥ 3.4 mmol/l for patients with multiple CVD risk factors; TGs < 5.6 mmol/l; maximally tolerated lipid-lowering therapy | Percent change in LDL-C | 12 weeks | Bempedoic acid 180 mg/day and ezetimibe 10 mg/day | 86 | 62.2 ± 9.5 | 44 (51.2%) | −36.2% |
| | | | | | | Ezetimibe 10 mg/day | 86 | 65.1 ± 8.4 | 43 (50.0%) | −23.2% |
| | | | | | | Bempedoic acid 180 mg/day | 88 | 65.2 ± 9.8 | 48 (54.5%) | −17.2% |
| | | | | | | Placebo | 41 | 65.4 ± 10.8 | 17 (41.5%) | +1.8% |
| NCT02659397 | Lalwani, 2019 [22] | Multicenter, randomized, double-blind, placebo-controlled, parallel-group, phase II clinical study | 18–70 years of age; BMI ≥ 18 kg/m$^2$ and ≤40 kg/m$^2$; no history of CVD; treatment with atorvastatin 80 mg/day | Percent change in LDL-C from baseline to week 4; fold change in $C_{max}$ from baseline to week 2; fold change in AUC from baseline to week 2 | 4 weeks | Bempedoic acid 180 mg/day | 45 | 58 (10)* | 21 (51.2%) | −13.3% |
| | | | | | | Placebo | 23 | 58 (8)* | 10 (43.5%) | +9.2% |
| CLEAR Serenity (NCT02988115) | Laufs, 2019 [9] | Multicenter, randomized, double-blind, placebo-controlled, parallel-group, phase III clinical study | Men and postmenopausal or surgically sterile women; ≥18 years of age; history of intolerance of ≥2 statins; LDL-C ≥ 3.4 mmol/l for primary prevention patients and ≥2.4 mmol/l for HeFH patients | Percent change in LDL-C from baseline to week 12 | 24 weeks | Bempedoic acid 180 mg/day | 234 | 65.2 ± 9.7 | 133 (56.8%) | −23.6%° |
| | | | | | | Placebo | 111 | 65.1 ± 9.2 | 61 (55%) | −1.3%° |
| CLEAR Harmony (NCT02666664) | Ray, 2019 [23] | Multicenter, randomized, double-blind, placebo-controlled, parallel-group, phase III clinical study | Men and postmenopausal or surgically sterile women; ≥18 years of age; high CV risk; maximally tolerated lipid-lowering therapy; LDL-C ≥ 1.8 mmol/l | Overall safety, assessed according to the incidence of adverse events and changes in safety laboratory variables | 52 weeks | Bempedoic acid 180 mg/day | 1,487 | 65.8 ± 9.1 | 389 (26.1%) | −12.6% |
| | | | | | | Placebo | 742 | 66.8 ± 8.6 | 213 (28.7%) | +1.1% |
| CLEAR Tranquility (NCT03001076) | Ballantyne, 2018 [24] | Multicenter, randomized, double-blind, placebo-controlled, parallel-group, phase III clinical study | ≥18 years of age; history of intolerance to statin; low-dose statin therapy or no statin therapy; LDL-C ≥ 2.4 mmol/l | Percent change in LDL-C | 12 weeks | Bempedoic acid 180 mg/day | 181 | 63.8 ± 10.8 | 109 (60.2%) | −23.5% |
| | | | | | | Placebo | 88 | 63.7 ± 11.3 | 56 (63.6%) | +5.2% |

(*Continued*)

**Table 1.** (Continued)

| Study | First author, year [reference] | Study design | Main inclusion criteria | Primary outcomes | Treatment duration | Study groups | Patients, n | Age (years), mean ± SD | Female, n (%) | Average change in LDL-C from baseline |
|---|---|---|---|---|---|---|---|---|---|---|
| NCT02072161 | Ballantyne, 2016 [25] | Multicenter, randomized, double-blind, placebo-controlled, parallel-group, phase IIb clinical study | 18–80 years of age; BMI ≥ 18 kg/m² and ≤45 kg/m²; statin therapy; LDL-C ≥ 3 mmol/l and ≤5.7 mmol/l; TGs ≤ 4.5 mmol/l | Percent change in LDL-C | 12 weeks | Bempedoic acid 180 mg/day | 45 | 57 ± 10 | 31 (69%) | −24.3% |
| | | | | | | Bempedoic acid 120 mg/day | 44 | 59 ± 9 | 26 (61%) | −17.3% |
| | | | | | | Placebo | 45 | 56 ± 10 | 22 (49%) | −4.2% |
| NCT01941836 | Thompson, 2016 [26] | Multicenter, randomized, double-blind, controlled, parallel-group, phase IIb clinical study | 18–80 years of age; LDL-C ≥ 3.4 mmol/l and ≤5.7 mmol/l; TGs ≤ 4.5 mmol/l; BMI ≥ 18 kg/m² and ≤45 kg/m² | Percent change in LDL-C | 12 weeks | Bempedoic acid 180 mg/day and ezetimibe 10 mg/day | 24 | 59 ± 9 | 13 (54.2%) | −48.2% |
| | | | | | | Bempedoic acid 120 mg/day and ezetimibe 10 mg/day | 26 | 59 ± 10 | 14 (54%) | −43.3% |
| | | | | | | Ezetimibe 10 mg/day | 99 | 60 ± 10 | 52 (51.5%) | −21.2% |
| NCT01607294 | Gutierrez, 2014 [27] | Single-center, randomized, double-blind, placebo-controlled, parallel-group, phase II clinical study | Type 2 diabetes; low risk for CVD; 18–70 years of age; LDL-C ≥ 2.4 mmol/l; BMI ≥ 25 kg/m² and ≤35 kg/m² | Percent change in LDL-C | 4 weeks | Bempedoic acid 80 mg/day for 2 weeks followed by bempedoic acid 120 mg/day for 2 weeks | 30 | 55.3 ± 6.9 | 13 (43.4%) | −42.9% |
| | | | | | | Placebo | 30 | 56.0 ± 9.9 | 10 (33.3%) | −4.3% |
| NCT01751984 | Thompson, 2015 [28] | Multicenter, randomized, double-blind, placebo-controlled, parallel-group, phase II clinical study | Men and postmenopausal or surgically sterile women; 18–80 years of age; history of intolerance ≥1 statin; LDL-C ≥ 2.4 mmol/l and ≤5.7 mmol/l; TGs < 4 mmol/l; BMI ≥ 18 kg/m² and ≤40 kg/m² | Percent change in LDL-C | 8 weeks | Bempedoic acid 60 mg/day for 2 weeks followed by increasing dose at 2-week intervals to 120, 180, and 240 mg/day | 37 | 64 ± 5 | 17 (46%) | −32.5% |
| | | | | | | Placebo | 19 | 60 ± 8 | 11 (58%) | −3.3% |

(*Continued*)

**Table 1.** (Continued)

| Study | First author, year [reference] | Study design | Main inclusion criteria | Primary outcomes | Treatment duration | Study groups | Patients, n | Age (years), mean ± SD | Female, n (%) | Average change in LDL-C from baseline |
|---|---|---|---|---|---|---|---|---|---|---|
| NCT01262638 | Ballantyne, 2013 [29] | Multicenter, randomized, double-blind, placebo-controlled, parallel-group, phase II clinical study | 18–80 years of age; LDL-C ≥ 3.4 mmol/l and ≤5.2 mmol/l; TGs < 4.5 mmol/l; BMI ≥ 18 kg/m² and ≤35 kg/m² | Percent change in LDL-C | 12 weeks | Bempedoic acid 120 mg/day | 44 | 57 ± 10 | 19 (43%) | −26.6% |
| | | | | | | Bempedoic acid 80 mg/day | 44 | 59 ± 9 | 21 (48%) | −25.4% |
| | | | | | | Bempedoic acid 40 mg/day | 45 | 58 ± 9 | 26 (58%) | −17.9% |
| | | | | | | Placebo | 44 | 56 ± 10 | 13 (30%) | −2.1% |

*Expressed as median (standard deviation).

°After 12 weeks of treatment.

ASCVD, atherosclerotic cardiovascular disease; AUC, area under the curve; BMI, body mass index; $C_{max}$, peak plasma concentration; CV, cardiovascular; CVD, cardiovascular disease; HeFH, heterozygous familial hypercholesterolemia; LDL-C, low-density lipoprotein cholesterol; TG, triglyceride.

bempedoic acid had a barely detectable significant effect on HDL-C particle number ($n = 271$; MD −3.21%; 95% CI −6.40%, −0.02%; $p = 0.049$; $I^2 = 43.3\%$) (Fig 9).

There were no significant effects on TGs ($n = 2,954$; MD −1.51%; 95% CI −3.75%, 0.74%; $p = 0.189$; $I^2 = 15.1\%$) (Fig 10), VLDL particle number ($n = 271$; MD 3.79%; 95% CI −9.81%, 17.39%; $p = 0.585$; $I^2 = 35.1\%$) (Fig 11), and Apo A-1 ($n = 382$; MD −1.83%; 95% CI −5.23%, 1.56%; $p = 0.290$; $I^2 = 50.1\%$) (Fig 12). When the largest study (the CLEAR Harmony trial) [23] was excluded from the meta-analysis, all the effect sizes were similar (S1 Table). Furthermore,

**Table 2. Risk of bias evaluation of the studies according to Cochrane guidelines.**

| First author, year [reference] | Sequence generation | Allocation concealment | Blinding of participants, personnel, and outcome assessment | Incomplete outcome data | Selective outcome reporting | Other potential threats to validity |
|---|---|---|---|---|---|---|
| Ballantyne, 2019 [21] | L | L | L | H | U | U |
| Lalwani, 2019 [22] | L | L | L | L | L | L |
| Laufs, 2019 [9] | L | L | L | L | L | L |
| Ray, 2019 [23] | L | L | L | L | L | L |
| Ballantyne, 2018 [24] | L | L | L | L | L | L |
| Ballantyne, 2016 [25] | L | L | L | L | L | L |
| Thompson, 2016 [26] | L | L | L | L | L | L |
| Gutierrez, 2014 [27] | L | L | L | L | L | L |
| Thompson, 2015 [28] | L | L | L | L | L | L |
| Ballantyne, 2013 [29] | L | L | L | L | L | L |

H, high risk of bias; L, low risk of bias; U, unclear risk of bias.

## Total Cholesterol

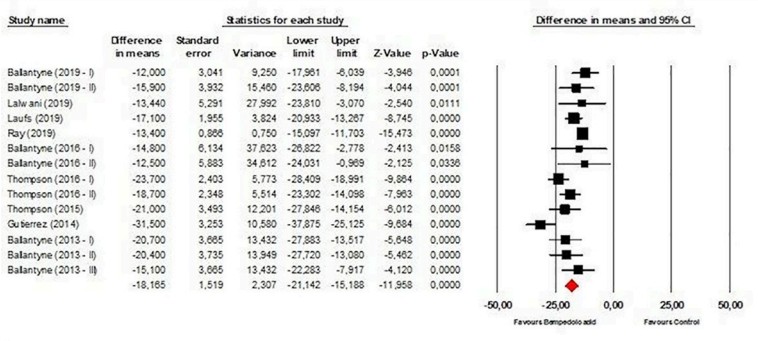

**Fig 2. Forest plot displaying mean difference and 95% confidence intervals for the effect of bempedoic acid on plasma levels of total cholesterol.**

the effect sizes were robust in the leave-one-out sensitivity analysis (S1–S4 Figs) and not mainly driven by a single study.

Visual inspection of Begg's funnel plots did not reveal any asymmetry, suggesting no publication bias for the effect of bempedoic acid on the investigated parameters (S6 Fig).

Duval and Tweedie's "trim and fill" method yielded 1 potentially missing study on the left side of the plot for TC, increasing the effect size to −15.27% (95% CI −17.61%, −12.92%); 4 potentially missing studies on the left side of the plot for HDL-C, lowering the effect size to −5.88% (95% CI −6.18%, −5.57%); 1 potentially missing study on the right side of the plot for HDL particle number, lowering the effect size to −1.86% (95% CI −4.86%, 1.13%); 4 potentially missing studies on the left side of the plot for non-HDL-C, increasing the effect size to −20.15% (95% CI −23.73%, −16.57%); 3 potentially missing studies on the left side of the plot for LDL-C, increasing the effect size to −25.17% (95% CI −29.55%, −20.79%); 2 potentially missing studies on the left side of the plot for LDL particle number, lowering the effect size to −21.85% (95% CI −24.74%, −18.96%); 1 potentially missing study on the right side of the funnel for VLDL particle number, increasing the effect size to 8.55% (95% CI −4.01%, 21.11%); 2 potentially missing studies on the left side of the plot for Apo A-1, lowering the effect size to −3.77% (95% CI −7.33%, −0.21%); and 3 potentially missing studies on the right side of the

## Non HDL-Cholesterol

**Fig 3. Forest plot displaying mean difference and 95% confidence intervals for the effect of bempedoic acid on plasma levels of non-high-density lipoprotein (HDL) cholesterol.**

## LDL-Cholesterol

**Fig 4. Forest plot displaying mean difference and 95% confidence intervals for the effect of bempedoic acid on plasma levels of low-density lipoprotein (LDL) cholesterol.**

plot for hsCRP, lowering the effect size to −25.69% (95% CI −29.89%, −21.48%). However, neither Begg's rank correlation nor Egger's linear regression confirmed the presence of publication bias for the analyses ($p > 0.05$ for all) (S2 Table).

The classic fail-safe $N$ test suggested that the following number of studies with negative results would be needed to bring the estimated effect size for each outcome to a non-significant level: 2,280 studies for TC ($p < 0.001$ for the test), 838 studies for HDL-C ($p < 0.001$ for the test), 2 studies for HDL particle number ($p = 0.017$ for the test), 2,004 studies for non-HDL-C ($p < 0.001$ for the test), 2,053 studies for LDL-C ($p < 0.001$ for the test), 263 studies for LDL particle number ($p < 0.001$ for the test), 1,308 studies for Apo B ($p < 0.001$ for the test), and 188 studies for hsCRP ($p < 0.001$ for the test). The individual analyses are included in S3 Table.

## Safety analysis

Bempedoic acid was positively associated with an increased risk of discontinuation of treatment ($n = 3,731$; OR 1.37; 95% CI 1.06, 1.76; $p = 0.015$; $I^2 = 0\%$), elevated SUA ($n = 569$; OR 3.55; 95% CI 1.03, 12.27; $p = 0.045$; $I^2 = 0\%$), elevated liver enzymes ($n = 2,363$; OR 4.28; 95% CI 1.34, 13.71; $p = 0.014$; $I^2 = 0\%$), and elevated CK ($n = 2,718$; OR 3.79; 95% CI 1.06, 13.51; $p = 0.04$; $I^2 = 0\%$), but it was strongly associated with a decreased risk of new onset or worsening diabetes ($n = 2,498$; OR 0.59; 95% CI 0.39, 0.90; $p = 0.01$; $I^2 = 0\%$) (Fig 13).

These findings were robust in the leave-one-out sensitivity analyses (S6 Fig). However, when the data from the largest study (the CLEAR Harmony trial) [23] were excluded from the

## LDL Particle Number

**Fig 5. Forest plot displaying mean difference and 95% confidence intervals for the effect of bempedoic acid on plasma levels of low-density lipoprotein (LDL) particle number.**

**Fig 6. Forest plot displaying mean difference and 95% confidence intervals for the effect of bempedoic acid on plasma levels of apolipoprotein B.**

meta-analysis, the effect sizes for the safety outcomes lost their statistical significance (S1 Table).

The incidence of the other AEs did not differ between groups (S4 Table). Considering the reasons for treatment discontinuation in included trials that reported all or part of them (S5 Table), it was not possible to identify the responsible reasons of the effect size of Fig 6 (S7 Fig).

Visually, the funnel plot of standard error by log OR was slightly asymmetric only for risk of discontinuation of treatment. This asymmetry was imputed to 6 potentially missing studies on the right side of the funnel plot, increasing the estimated risk to 1.55 (95% CI 1.22, 1.97) (S8 Fig). The presence of publication bias for the analysis was confirmed by Egger's linear regression ($p = 0.005$), but not by Begg's rank correlation ($p = 0.298$). The classic fail-safe $N$ test suggested that 1 study with a negative result would be needed to bring the estimated risk of CK elevation to a non-significant level ($p = 0.042$ for the test), and 2 studies with negative results would be needed to bring the estimated risk of transaminase elevation to a non-significant level ($p = 0.021$ for the test). The individual analyses are included in S6 Table.

## Discussion

Inhibitors of 3-hydroxy-3-methylglutaryl coenzyme A reductase (statins) represent the first-line treatment for dyslipidemia, being able to reduce LDL-C by 30%–50% and subsequently

**Fig 7. Forest plot displaying mean difference and 95% confidence intervals for the effect of bempedoic acid on plasma levels of high-density lipoprotein (HDL) cholesterol.**

## High sensitivity-C Reactive Protein

| Study name | Statistics for each study | | | | | | | Difference in means and 95% CI |
| --- | --- | --- | --- | --- | --- | --- | --- | --- |
| | Difference in means | Standard error | Variance | Lower limit | Upper limit | Z-Value | p-Value | |
| Lalwani (2019) | -35,335 | 21,555 | 464,619 | -77,582 | 6,912 | -1,639 | 0,1012 | |
| Laufs (2019) | -29,500 | 6,255 | 39,121 | -41,759 | -17,241 | -4,716 | 0,0000 | |
| Ray (2019) | -25,000 | 2,646 | 7,001 | -30,186 | -19,814 | -9,449 | 0,0000 | |
| Ballantyne (2016 - I) | -29,800 | 12,743 | 162,379 | -54,775 | -4,825 | -2,339 | 0,0194 | |
| Ballantyne (2016 - II) | -21,800 | 12,463 | 155,319 | -46,226 | 2,626 | -1,749 | 0,0803 | |
| Thompson (2016 - I) | -36,100 | 10,136 | 102,733 | -55,966 | -16,234 | -3,562 | 0,0004 | |
| Thompson (2016 - II) | -48,600 | 12,511 | 156,533 | -73,122 | -24,078 | -3,884 | 0,0001 | |
| | -27,033 | 2,240 | 5,018 | -31,423 | -22,642 | -12,067 | 0,0000 | |

Favours Bempedoic acid — Favours Control

**Fig 8. Forest plot displaying mean difference and 95% confidence intervals for the effect of bempedoic acid on plasma levels of high-sensitivity C-reactive protein.**

decrease the incidence of CV events [30]. Despite the highly favorable benefit/risk profile of statins, a large number of patients are statin intolerant or need additional lipid-lowering drugs to reach optimal LDL-C levels [3]. The current meta-analysis shows that bempedoic acid safely reduces LDL-C levels by about 23%, suggesting that it might be considered as an effective alternative or add-on therapy to statins or ezetimibe.

About 31%–49% or more of patients with hyperlipidemia do not achieve LDL-C goals with current lipid-lowering therapies [31,32], and more than half of patients stop statin treatment within 1 year of initiation [33]. Sixty percent of patients who discontinue statins report different symptoms of drug intolerance as the main reason for discontinuation [34]. Statin intolerance, usually characterized by myalgia, myositis, and/or myopathy, occurs in 2%–15% of users, the estimate being strongly variable in epidemiological and rechallenging studies [35,36]. Furthermore, large meta-analyses showed that statin treatment is associated with a 9%–13% increase in risk of developing diabetes [37]. However, scientifically unsupported concerns about statin safety spread by mass media lead to the formation of a negative image of these drugs and increase of their cessation rate [38].

Additional treatments of dyslipidemia include ezetimibe (second-line) and fenofibrate (third-line). Ezetimibe, in combination with statin therapy, lowers LDL-C by an additional 20% or so [39] and significantly reduces the risk of major adverse CV events, non-fatal myocardial infarction, and non-fatal stroke compared with statins alone, with less or no effect on fatal endpoints [40]. A simulation based on adding ezetimibe in a huge statin-treated cohort suggests that the percentage of patients with LDL-C > 1.8 mmol/l and >2.4 mmol/l would fall from 65% to 38% and from 25% to 12%, respectively [41].

## HDL Particle Number

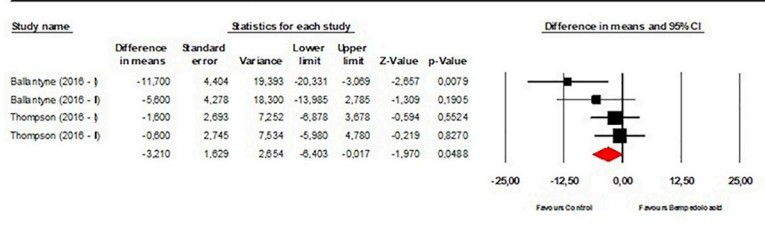

| Study name | Statistics for each study | | | | | | | Difference in means and 95% CI |
| --- | --- | --- | --- | --- | --- | --- | --- | --- |
| | Difference in means | Standard error | Variance | Lower limit | Upper limit | Z-Value | p-Value | |
| Ballantyne (2016 - I) | -11,700 | 4,404 | 19,393 | -20,331 | -3,069 | -2,657 | 0,0079 | |
| Ballantyne (2016 - II) | -5,600 | 4,278 | 18,300 | -13,985 | 2,785 | -1,309 | 0,1905 | |
| Thompson (2016 - I) | -1,600 | 2,693 | 7,252 | -6,878 | 3,678 | -0,594 | 0,5524 | |
| Thompson (2016 - II) | -0,600 | 2,745 | 7,534 | -5,980 | 4,780 | -0,219 | 0,8270 | |
| | -3,210 | 1,629 | 2,654 | -6,403 | -0,017 | -1,970 | 0,0488 | |

Favours Control — Favours Bempedoic acid

**Fig 9. Forest plot displaying mean difference and 95% confidence intervals for the effect of bempedoic acid on plasma levels of high-density lipoprotein (HDL) particle number.**

## Triglycerides

| Study name | Difference in means | Standard error | Variance | Lower limit | Upper limit | Z-Value | p-Value | Difference in means and 95% CI |
|---|---|---|---|---|---|---|---|---|
| Lalwani (2019) | 8,270 | 9,733 | 94,734 | -10,807 | 27,347 | 0,850 | 0,3955 | |
| Laufs (2019) | 0,500 | 4,595 | 21,118 | -8,507 | 9,507 | 0,109 | 0,9134 | |
| Ray (2019) | -1,370 | 1,287 | 1,655 | -3,892 | 1,152 | -1,065 | 0,2870 | |
| Ballantyne (2016 - I) | -6,100 | 8,528 | 72,726 | -22,814 | 10,614 | -0,715 | 0,4744 | |
| Ballantyne (2016 - II) | -1,800 | 6,219 | 38,682 | -13,990 | 10,390 | -0,289 | 0,7723 | |
| Thompson (2016 - I) | -5,200 | 6,729 | 45,274 | -18,388 | 7,988 | -0,773 | 0,4396 | |
| Thompson (2016 - II) | -11,900 | 5,935 | 35,221 | -23,532 | -0,268 | -2,005 | 0,0449 | |
| Thompson (2015) | 18,600 | 10,948 | 119,864 | -2,858 | 40,058 | 1,699 | 0,0893 | |
| | -1,505 | 1,147 | 1,315 | -3,753 | 0,742 | -1,313 | 0,1893 | |

**Fig 10. Forest plot displaying mean difference and 95% confidence intervals for the effect of bempedoic acid on plasma levels of triglycerides.**

Fibrates are less effective on LDL-C levels, with their main indication being moderate-to-severe hypertriglyceridemia, such that they are rarely used in cardiology settings. However, a large meta-analysis of 16,112 patients showed evidence for a protective effect compared to placebo for the primary composite outcome of non-fatal myocardial infarction, non-fatal stroke, and vascular death [42]. Besides, patients with very high baseline LDL-C level or very high or extreme global CV risk need additional lipid-lowering drugs to optimize the lipid profile [43], especially in light of the most recent international recommendations [44,45]. Monoclonal antibodies that target proprotein convertase subtilisin/kexin type 9 (PCSK9) have recently been demonstrated to dramatically reduce LDL-C level (even over 60%) in the majority of cases, while significantly reducing CV risk; however, their cost–benefit ratio is yet under discussion, and in many countries their use is limited due to strict reimbursement rules [46]. In this context, there is yet place for the development of new less-expensive, effective/safe lipid-lowering drugs.

By analyzing data from 10 phase II and phase III RCTs including a total of 3,788 patients, we confirmed that bempedoic acid significantly reduced TC (by 15%), non-HDL-C (by 18.2%), LDL-C (by 22.9%), LDL particle number (by 20.7%), Apo B (by 15.2%), and hsCRP (by 27%), while negatively affecting serum levels of HDL-C (−5.8%) and HDL particle number (−3.2%). These findings strengthen the unpowered data previously reported by Wang et al., based on only 625 patients [47]. These findings could also be quantitatively relevant, since they have usually been obtained when bempedoic acid is administered on top of an effective lipid-lowering treatment, with a quite good safety and tolerability profile.

Our results also confirmed that bempedoic acid therapy is overall safe and well tolerated, with no significant increase of serious AEs. However, an increase of drug discontinuation and

## VLDL Particle Number

| Study name | Difference in means | Standard error | Variance | Lower limit | Upper limit | Z-Value | p-Value | Difference in means and 95% CI |
|---|---|---|---|---|---|---|---|---|
| Ballantyne (2016 - I) | -19,200 | 16,755 | 280,723 | -52,039 | 13,639 | -1,146 | 0,2518 | |
| Ballantyne (2016 - II) | -0,900 | 13,940 | 194,316 | -28,221 | 26,421 | -0,065 | 0,9485 | |
| Thompson (2016 - I) | 24,600 | 12,967 | 168,139 | -0,815 | 50,015 | 1,897 | 0,0578 | |
| Thompson (2016 - II) | 0,900 | 12,781 | 163,349 | -24,150 | 25,950 | 0,070 | 0,9439 | |
| | 3,792 | 6,937 | 48,128 | -9,805 | 17,389 | 0,547 | 0,5847 | |

**Fig 11. Forest plot displaying mean difference and 95% confidence intervals for the effect of bempedoic acid on plasma levels of very-low-density lipoprotein (VLDL) particle number.**

## Apolipoprotein A-1

| Study name | Statistics for each study | | | | | | | Difference in means and 95% CI |
|---|---|---|---|---|---|---|---|---|
| | Difference in means | Standard error | Variance | Lower limit | Upper limit | Z-Value | p-Value | |
| Lalwani (2019) | 2,020 | 3,403 | 11,581 | -4,650 | 8,690 | 0,594 | 0,5528 | |
| Ballantyne (2016 - I) | 3,600 | 3,460 | 11,972 | -3,182 | 10,382 | 1,040 | 0,2981 | |
| Ballantyne (2016 - II) | 1,700 | 3,335 | 11,120 | -4,836 | 8,236 | 0,510 | 0,6102 | |
| Thompson (2016 - I) | -6,100 | 2,293 | 5,257 | -10,594 | -1,606 | -2,661 | 0,0078 | |
| Thompson (2016 - II) | -4,800 | 2,295 | 5,268 | -9,299 | -0,301 | -2,091 | 0,0365 | |
| Thompson (2015) | -4,300 | 3,553 | 12,622 | -11,263 | 2,663 | -1,210 | 0,2261 | |
| | -1,833 | 1,732 | 3,001 | -5,228 | 1,563 | -1,058 | 0,2901 | |

-25,00   -12,50   0,00   12,50   25,00

Favours Control        Favours Bempedoic acid

**Fig 12. Forest plot displaying mean difference and 95% confidence intervals for the effect of bempedoic acid on plasma levels of apolipoprotein A-1.**

elevations of SUA, transaminase, and CK were observed. The detailed analysis of the reasons for discontinuation (see S5 Table) reported in the available trials does not give any clear pattern that could explain the 37% increased risk of discontinuation of bempedoic acid in comparison to placebo; this issue, however, needs to be further investigated. As for the other adverse effects possibly related to bempedoic acid, it is important to emphasize that in the 4 trials where CK increase was reported, it was observed in only 16 patients (of 1,792 investigated; 0.9%), and only single patients had a repeated and confirmed CK elevation greater than 5 times the upper limit of normal. More data with longer follow-up are also necessary to confirm the risk of SUA increase with bempedoic acid (observed in only 3 trials, where SUA increase was observed in 18/354 [5%]), as well as the risk of transaminase increase (observed in 5 trials, where transaminase increase was observed in only 1.1% of patients [20/1,823] in active-treated group). It is also worth emphasizing that bempedoic acid, due to its mechanism of action, does not increase the risk muscle-related side-effects and significantly reduces the risk of worsening or new onset diabetes by about 40% (however, based only on 2 available studies)—AEs that might be relatively often observed in statin trials, especially for high- and very-high-risk patients requiring intense therapy.

In this context, bempedoic acid seems to be an interesting option as an overall safe drug to be easily associated to statins and ezetimibe. In particular, the drug will be marketed as monotherapy or in a single pill with ezetimibe for the management of statin-intolerant patients. Considering the different mechanism of action of bempedoic acid and ezetimibe, the high safety profile of both drugs, and the lack of interaction risk between them, it is expected that this association will be a relatively effective and safe lipid-lowering treatment.

The main limitation of this meta-analysis is related to the relatively small number of patients involved in the studies, which were often short or middle term, as well as their heterogeneity (including different populations that were investigated, i.e., patients with type 2 diabetes, hypercholesterolemia, or statin intolerance). Moreover, heterogeneity of effects is moderate to large across most of the biochemical outcomes. Data on decreased CV events and mortality are lacking for bempedoic acid as well [48].

In conclusion, the current meta-analysis demonstrates an acceptable safety profile and multiple positive effects of bempedoic acid on lipid profile and hsCRP serum levels.

Further data on the cost–benefit efficacy of bempedoic acid treatment will come from the CLEAR Outcomes study, a phase III, event-driven, randomised, multicenter, double-blind, placebo-controlled trial designed to evaluate whether treatment with bempedoic acid reduces the risk of CV events. The primary endpoint of the study is the effect of bempedoic acid on major adverse CV events (CV death, non-fatal myocardial infarction, non-fatal stroke, and coronary revascularization). The enrollment ended in November 2019 [49].

## Discontinuation of treatment

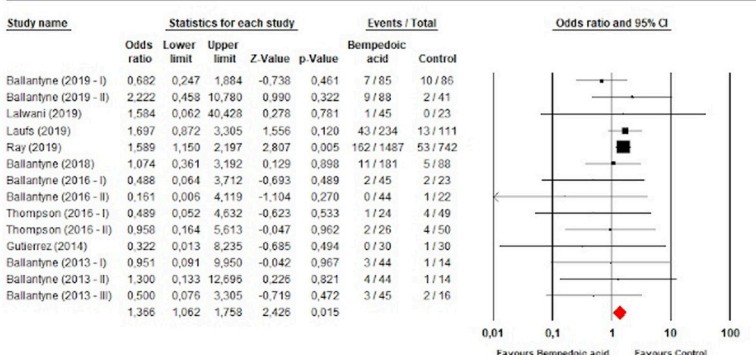

## Liver enzymes elevation

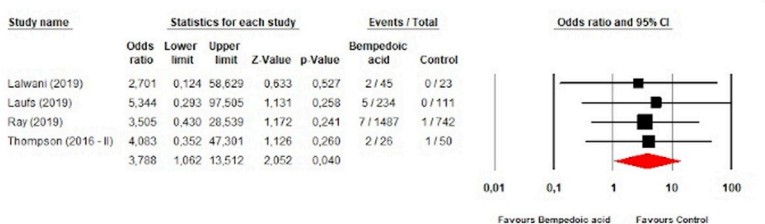

## Creatine kinase elevation

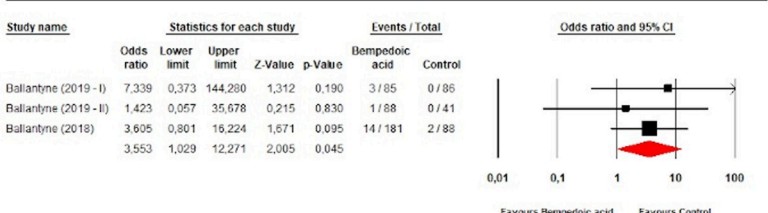

## Serum uric acid elevation

## New onset or worsening diabetes

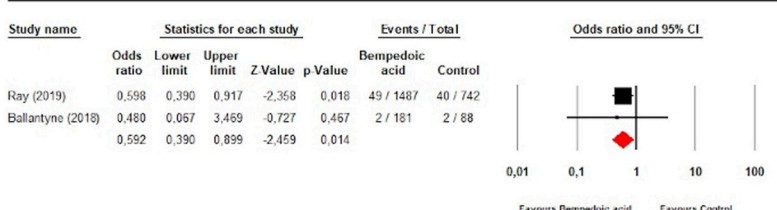

**Fig 13. Forest plot comparing the risk of adverse events statistically associated with bempedoic acid treatment.**

## Supporting information

**S1 PRISMA Checklist. PRISMA Checklist.**
(DOC)

**S1 Data. Summary data for all included studies.**
(XLS)

**S1 Fig. Forest plots showing leave-one-out for TC, non-HDL-C, and TG.**
(TIF)

**S2 Fig. Forest plots showing leave-one-out for LDL-C, LDL particle number, VLDL particle number, and Apo B.**
(TIF)

**S3 Fig. Forest plots showing leave-one-out for HDL-C, HDL particle number and Apo A-1.**
(TIF)

**S4 Fig. Forest plots showing leave-one-out for hsCRP.**
(TIF)

**S5 Fig. Funnel plots detailing publication bias in the studies reporting the effect of ETC-1002 treatment on serum lipids and hsCRP concentrations.**
(TIF)

**S6 Fig. Plot showing leave-one-out sensitivity analysis for safety analysis.**
(TIF)

**S7 Fig. Plot showing reasons for discontinuation to treatment as reported in the studies.**
*Data referring to statin-intolerant patients; $ Data referring to statin-tolerant patients.
(TIF)

**S8 Fig. Funnel plot detailing publication bias in the safety analysis.**
(TIF)

**S1 Table. Meta-analysis' findings after excluding the CLEAR Harmony study.**
(DOC)

**S2 Table. Begg's rank correlation nor Egger's linear regression tests.**
(DOC)

**S3 Table. Classic fail-safe N results for the efficacy analyses.**
(DOC)

**S4 Table. Adverse events occurred in at least 2 clinical trials.** AEs = Adverse events.
(DOC)

**S5 Table. Reasons of discontinuation to treatments as reported by the studies.**
(DOC)

**S6 Table. Classic fail-safe N results for the safety analyses.**
(DOC)

## Acknowledgments

The meta-analysis was prepared within the Lipid and Blood Pressure Meta-Analysis Collaboration (LBPMC) Group (http://www.lbpmcgroup.umed.pl).

The members of the International Lipid Expert Panel (ILEP) are F. Alnouri, F. Amar, A. G. Atanasov, G. Bajraktari, M. Banach, M. A. Bartlomiejczyk, B. Bjelakovic, E. Bruckert, A. Bielecka-Dabrowa, A. Cafferata, R. Ceska, A. F. G. Cicero, X. Collet, O. Descamps, N. Devaki, D. Djuric, R. Durst, M. V. Ezhov, Z. Fras, D. Gaita, S. von Haehling, A. V. Hernandez, S. R. Jones, J. Jozwiak, N. Kakauridze, N. Katsiki, A. Khera, K. Kostner, R. Kubilius, G. Latkovskis, G. B. J. Mancini, A. D. Marais, S. S. Martin, J. A. Martinez, M. Mazidi, D. P. Mikhailidis, E. Mirrakhimov, A. R. Miserez, O. Mitchenko, P. Moriarty, S. M. Nabavi, D. B. Panagiotakos, G. Paragh, D. Pella, P. E. Penson, Z. Petrulioniene, M. Pirro, A. Postadzhiyan, R. Puri, A. Reda, Ž. Reiner, J. Riadh, D. Richter, M. Rizzo, M. Ruscica, A. Sahebkar, N. Sattar, M. C. Serban, A. M. A. Shehab, A. B. Shek, C. R. Sirtori, C. Stefanutti, T. Tomasik, P. P. Toth, M. Viigimaa, D. Vinereanu, B. Vohnout, M. Vrablik, N. D. Wong, H. I. Yeh, J. Zhisheng, and A. Zirlik.

## Author Contributions

**Conceptualization:** Arrigo F. G. Cicero, Federica Fogacci, Maciej Banach.

**Data curation:** Arrigo F. G. Cicero, Federica Fogacci, Adrian V. Hernandez.

**Formal analysis:** Federica Fogacci.

**Methodology:** Adrian V. Hernandez.

**Project administration:** Maciej Banach.

**Writing – original draft:** Arrigo F. G. Cicero, Federica Fogacci, Maciej Banach.

**Writing – review & editing:** Arrigo F. G. Cicero, Federica Fogacci, Adrian V. Hernandez, Maciej Banach.

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
