## [Decision Letter · Decision Letter 0]

24 Dec 2019

Dear Dr. Banach,

Thank you very much for submitting your manuscript "Efficacy and safety of bempedoic acid: a systematic review and meta-analysis of phase 2 and 3 randomized controlled trials" (PMEDICINE-D-19-03833) for consideration at PLOS Medicine. 

[LINK]

In light of these reviews, I am afraid that we will not be able to accept the manuscript for publication in the journal in its current form, but we would like to consider a revised version that addresses the reviewers' and editors' comments. Obviously we cannot make any decision about publication until we have seen the revised manuscript and your response, and we plan to seek re-review by one or more of the reviewers. 

We expect to receive your revised manuscript by Jan 14 2020 11:59PM. Please email us (plosmedicine@plos.org) if you have any questions or concerns.

We look forward to receiving your revised manuscript. 

Sincerely,

Adya Misra, PhD

Senior Editor 

PLOS Medicine

plosmedicine.org

Title- please include a descriptor for bempedoic acid in the title such as “lipid lowering drug” or similar

Abstract- in the background please provide a sentence about bempedoic acid use in humans in more accessible language. For example- has it been used to reduce LDL-c etc. 

Abstract- methods and findings sections must be combined in PLOS Medicine style and the last sentence in this section should outline the limitations of your methodology

Abstract-please provide a sentence about the search, including dates and databases searched. 

Abstract- please correct the typo in “mean differences”

Abstract- please provide 95% CI and p values for all the results reported in the abstract, not only the positive findings 

Abstract- since you have investigated and reported on safety in the main manuscript, please include these findings in the abstract for greater transparency- in the methods, findings and conclusions sections

References- please use Vancouver style and provide references within square brackets throughout 

Line 78- please revise to be more specific “showing and overall interesting impact of bempedoic acid on human lipid pattern”

Line 110- please elaborate “Studies that lacked a properly controlled design for bempedoic acid treatment were excluded”

Please revise “diabetics” to patients with type 2 diabetes. The same goes for hypercholestrolemics

Table 2- please provide annotations for all letters, L H and U. 

Results- the effect on HDL particle number (p=0.049) cannot be described as “significant” along with those that were p<0.01. Please provide this result separately as it is so close to p=0.05

Please include a separate paragraph for the analysis on missing studies and the lowering of effect sizes. Please provide individual p values here along with the confidence intervals. 

Line 243- “However, neither Begg’s rank correlation nor Egger’s linear

regression confirmed the presence of publication for the analyses (p>0·05 always)” please provide these analyses as supplementary information and please provide the exact p value instead of p>0.05

Please provide exact p values for the results on N tests and provide the individual analyses as supplementary information 

Figure 1- several p values are p=0.000, please correct this to an appropriate p value of either p<0.001 or provide the exact p values if p>0.001. As pointed out by Reviewer 1, each meta-analysis should be provided in an individual figure for more precise plotting. You may choose to include some of these as supplementary figures as you see fit. 

Figure 1 should be a flowchart of records screened in this study

Results- please include a sentence about the degree of heterogeneity observed and what was done to identify the sources of the heterogeneity. 

Discussion- this sentence requires copyediting and a reference to support “Despite statins should be regarded as one of the major advances of modern preventive medicine, a large number of patients needs additional lipid lowering to reach the optimal LDL-C levels or is statin intolerant”

Discussion- it is unclear if this refers to the current submission? If so, this was not specifically studies in the SR/MA and should be removed “We showed that bempedoic acid might be such an effective alternative as an add-on to statins or ezetimibe or in monotherapy as it might reduce LDL-C levels by about 23% and inflammation level (hsCRP) by over 27% with acceptable overall tolerance”

Discussion focusses on statins and other treatments of dyslipidemia which should be provided in the introduction as a condensed paragraph. Please present and organize the Discussion as follows: a short, clear summary of the article's findings; what the study adds to existing research and where and why the results may differ from previous research; strengths and limitations of the study; implications and next steps for research, clinical practice, and/or public policy; one-paragraph conclusion.

Discussion- “Our results also confirmed that this therapy is overall safe and well tolerated, with no significant increase of serious adverse effects” this sentence is not supported by the results and must be revised to fully acknowledge the risk of discontinuation of treatment and other AEs as needed

Discussion- muscle related side effects were not studied in this SR/MA and therefore this sentence should be removed “It is also worth emphasizing that bempedoic acid, due to its mechanism of action, does not increase the risk muscle-related side effects and significantly reduce the risk of worsening or new onset diabetes by about 40% (however based only on two available studies), what might be relatively often observed in statin trials, especially for high and very high risk patients requiring intense therapy”

Discussion- a number of trials were short term or have not yet been completed. This should be included as a limitation more clearly 

PRISMA checklist- please remove page numbers as these are likely to change and include sections or paragraphs 

Supplementary information- Table S3 contains adverse events “general disorders” please include a more specific description. 

Comments from the reviewers:

Reviewer #1: I confine my remarks to statisitical aspects of this paper

I have only a couple of minor points to fix before I can recommend publication.

Line 47 Insert "significant" between not and modify

Line 56 Instead of (or in addition to) ranks of causes, give the number dead or rate of death. Ranks have two problems in this context: 1) Something has to be the leading cause and 2) Ranks depend on how causes are divided up (is "cancer" one thing or many?)

Line 152 This is not the right format. It should be use the square root symbol, no parens is needed around the first two terms, the subscripts could be pre and post for brevity. 

Line 153: On what basis was R = 0.5 chosen?

Table 2: What about H and U (in subscript)?

Figures: I can see why you did them this way, but I'd consdier a figure for each MA (so, figure 1 would be 3 figures); this would let you change the minima and maxima of the x-axis to allow more precise plotting. That would be 15 figures altogether, but the same ones, really. I'm not sure if that's better but I think it is. A simpler alterntative is to print them landscape instead of portrait

Peter Flom

Reviewer #2: The "Efficacy and safety of bempedoic acid: a systematic review and meta-analysis of phase 2 and 3 randomized controlled trials" is a systematic review and meta-analysis of ten studies, published between the years 2013 and 2019, including a total of 3788 subjects, with 2460 in the active-treated arm and 1328 in the control one. The main endpoint was the effect of bempedoic acid on lipid profile and high-sensitivity C-reactive protein (hsCRP) serum concentrations. For safety analyses, odds ratios (OR) and 95%CI were calculated using the Mantel-Haenszel method. According to this work, bempedoic acid significantly reduced total colesterol, non high-density lipoprotein cholesterol, low-density lipoprotein cholesterol, LDL particle number, apolipoprotein B, HDL-cholesterol, and HDL-particle number. Bempedoic acid failed to modify triglycerides levels, very-low-density lipoprotein particle number and Apolipoprotein A-1. Secondary effects and possible causes of discontinuations were also exposed. The authors attempted a very ambitious work combining relevant literature regarding a novel drug that might play a significant role in dyslipidemia treatment in the future. The article shows several strenghts such as modern clinical relavance, an extended description of the statistical analysis, and that it serves as a generator of hypothesis for future inverstigations. The article structure should be improved along with clarification of some statements supporting results with scarce evidence. 

 A clearer structure separating main ideas by paragraphs are recommended highliting and separating the endpoints of efficacy and safety. The results concerning lipid profiles, hsCRP, are mixed and lost within the paragraphs, blended with other secondary endpoints. 

 According to the journal recommendations, systematic reviews registered in the PROSPERO database should provide the registry number in their abstract. State all authors´emails on the manuscript as well as author contributions. According to the journal publication guidelines, the corresponding author should have an ORCID ID adressed in the manuscript. 

 The writers are asked to indicate, within the methods section, if a review protocol exists, if and where it can be accessed, and, if so, provide a copy of the protocol as Supporting Information. Alternatively, the authors should include more information that clarifies and justifies their choice of methods. In line 121, the "adverse events" must be clearly stated, as so should be the "major cardiac adverse events" (death, heart failure...?) in line 122. When abstracting quantitative measures such as worsening diabetes or descrease in glomerular filtration rate, please describe how much change in levels is threshhold to categorize as an event. 

 With respect to the baseline characteristics of the evealuated studies summarized in table 1, the authors are off to a good start, however, it might be important to expose the main endpoints and results of the different studies. 

 In the results it might be usefull to highlight as to why exclude form the meta-analysis the data from the largest study (line 223, line 283). 

 With respect to the results shown in Table S2, there is a lot of data and the table fails to be comprehensible. May be pick those events with the most biological plausability. 

 While the discussion appears to be sound, the prioritization of ideas is unclear for it should start with a brief summary of the main findings, following journal recommendations, giving the reader a quick glance of the main take-home message. The discussion should give a concise insight and be tightly argued, supporting the main results of the metaanalysis with previous litarature, which is not elaborated in the discussion. 

 Only two comments of the effect of acid bempedoic are written about the hsCPR, mixed with the results concerning the lipid profile; a separate paragraph might emphasize the importance of this finding. A clarification as to if secondary causes of change in CPR levels where excluded is advised. 

 It would be interesting to explain the etiology of the CK increase in patients with bempedoic acid for it does not act on muscle tissue enzime pathways (lines 353-356 and later 361-362). 

 The authors aim to demonstrate an effect of bempedoic acid on diabetes, however, the data does not fully support this conclusion, specifically supported by two studies with biases of their own (figure 5, line 362). 

 Please explain the phrase "The detailed analysis of the discontinuation reasons reported in the available trials does not give any clear pattern, which could explain 37% risk increase of discontinuation of bempedoic acid in comparison to placebo" (line 350-352). 

 The authors should revise the language to improve readability, with emphasis in the discussion. 

Reviewer #3: Standard meta-analysis of phase 2 & 3 studies of bempedoic acid for both efficacy and safety. Uses standard methodology with a systematic review and complete data synthesis. Early stage analysis given absence of significant additional data from on-going phase 3 studies and end-point studies.

[LINK]

---

## [Decision Letter · Decision Letter 1]

6 Apr 2020

Dear Dr. Banach,

Thank you very much for re-submitting your manuscript "Efficacy and safety of bempedoic acid for the treatment of hypercholesterolemia:

a systematic review and meta-analysis of phase 2 and 3 randomized controlled trials" (PMEDICINE-D-19-03833R1) for review by PLOS Medicine.

I have discussed the paper with my colleagues and the academic editor and it was also seen again by xxx reviewers. I am pleased to say that provided the remaining editorial and production issues are dealt with we are planning to accept the paper for publication in the journal.

[LINK]

We look forward to receiving the revised manuscript by Apr 13 2020 11:59PM. 

Sincerely,

Adya Misra, PhD

Senior Editor 

PLOS Medicine

plosmedicine.org

Requests from Editors:

Title- Please revise title to “Efficacy and safety of bempedoic acid for the treatment of hypercholesterolemia: a systematic review and meta-analysis” 

Abstract

Methods and findings- please move this sentence earlier in the section “We included 10 RCTs (n=3788) comprising 26 arms [active arm (n=2460); control arm (n=1328)]”.

Exact p-values must be provided unless p<0.001

Conclusion- please begin with “our results show” or similar 

Author summary

The section “why was this study done” does not clearly outline the need for this meta-analysis in plain language. Please add a sentence or two about why LDL-C treatment is important, for instance. 

Please also revise “is hardly to achieve” to “is hard to achieve”

Data availability

The Data Availability Statement (DAS) requires revision. For each data source used in your study: 

References- please place the full stop after the square brackets

Line 105- “so that” should be revised to “such that”

Methods

Please provide detailed search terms used along with limiters such that the search may be replicated. This can be provided as an SI file. Please reference the full list of search terms in SI file xxx 

Please also add a sentence noting that this manuscript is reported according to PRISMA guidelines and the checklist has been provided as SI file xx

Results

Line 371 onwards please provide exact p values, p<0.05 is not permitted

Line 413 should be revised to “…patients need additional lipid…”

Line 423 ought to say “criticism of statins in the media…”

Please ensure that you provide a clean version of the manuscript along with one that has tracked changes. The current PDF contains highlighted text which should be un-highlighted. 

Comments from Reviewers:

Reviewer #1: The authors have addressed my concerns and I now recommend publication

Peter Flom

Reviewer #3: The authors have revised many aspecst of the manuscript satisfactorily.

The authors downplay the effectiveness, safety and significance of ezetimibe therapy. This is the second line after statins for treatment of LDL-C in all major guidelines (USA, Europe etc). Fibrates are very much a third line therapy and have minimal use in cardiology services being more comonly used in the management of hypertriglyceridemia. Indeed a formualtion bempedoic acid with ezetimibe is proposed for marketing. This section needs to be re-written and the safety and efficacy of the combination described.

Statin intolerance occurs in 1-2% of users in systematic rechallenge studies. The quoted figure of 20-30% was challenged and discredited by both US and European Society consensus groups on statin myopathy and intolerance as well as in reviews of statin tolerability. Indeed in the PCSK-9 statin intolerance studies about 15% of patients recruited on the basis of intolerance to 3 statins recorded myalgia on placebo treatment.

The authors state that 30% of patients did not achieve the 2009 US targets (LTAP study) but do not provide more up to date US or European data. There have been 2 iterations of guidelines since that date. The international DYSIS-2 (Gitt AK et al; Atherosclerosis. 2016 Dec;255:200 (n=57885 patints) and Ferriere J et al; Eur J Prev Cardiol. 2018 Dec;25(18):1966 ) study suggests 31-49% LDL-C target attainment in 2018. A simulation based on adding ezetimibe in this cohort suggests that the percentage of patients with LDL-C >1.8mmol/L (70mg/dl) and >2.4mmol/L (100 mg/dl) would fall from 65 to 38% and from 25 to 12%, respectively (De Ferrari GM et al. J Cardiovasc Med. 2018 Sep;19(9):485). 

The statement about outcome trials is incorrect as is the statement about phase 3 studies. The authors should comment on the phase 3 CLEAR programme of trials on bempedoic acid https://www.esperion.com/pipeline/clinical-trials/ or through clinicaltrials.gov (with trial identifiers) which include descriptions of the 5 CLEAR trials comprising 3623 patients which follow standard protocols for phase 3 designs in the field of lipid-lowering established during the statin studies. Most of those trials are included in this meta-analysis (n=3543). Details of the CLEAR-outcomes study in phase 4 required for regulatory approval are available via the Esperion site and through clinicaltrials.gov. This study is now underway.

[LINK]

---

## [Editor Report · Decision Letter 2]

9 Jun 2020

Dear Prof. Banach, 

On behalf of my colleagues and the academic editor, Dr. Anthony Wierzbicki, I am delighted to inform you that your manuscript entitled "Efficacy and safety of bempedoic acid for the treatment of hypercholesterolemia:

a systematic review and meta-analysis" (PMEDICINE-D-19-03833R2) has been accepted for publication in PLOS Medicine. 

PRODUCTION PROCESS

PRESS

PROFILE INFORMATION

Thank you again for submitting the manuscript to PLOS Medicine. We look forward to publishing it. 

Best wishes, 

Adya Misra, PhD

Senior Editor 

PLOS Medicine

plosmedicine.org